# Diagnosis and Treatment of Adenomyosis with Office Hysteroscopy—A Narrative Review of Literature

**DOI:** 10.3390/diagnostics13132182

**Published:** 2023-06-27

**Authors:** Fani Gkrozou, Anastasia Vatopoulou, Chara Skentou, Minas Paschopoulos

**Affiliations:** Department of Obstetrics and Gynaecology, Medical School, University of Ioannina, 45500 Ioannina, Greece; anastasiavat@hotmail.com (A.V.); haraskentou@gmail.com (C.S.); mpaschop@gmail.com (M.P.)

**Keywords:** adenomyosis, hysteroscopy, vaginoscopic approach, management, treatment, diagnosis

## Abstract

Introduction: Adenomyosis is a common chronic disease in women of reproductive age, characterised by the presence of ectopic endometrial tissue within myometrium. Even though adenomyosis presents with chronic pelvic pain, menorrhagia or abnormal uterine bleeding, dysmenorrhoea, and dyspareunia and is often recognised after hysterectomies. However, the development of ultrasonography and magnetic resonance imaging has improved the pre-operative diagnosis of the disease. Hysteroscopy provides information in real time from the uterine cavity and the offers the possibility of obtaining direct biopsies. Material and Methods: The literature was searched via Pubmed and Embase with the following headings: diagnosis of adenomyosis or adenomyoma and office hysteroscopy, hysteroscopy findings of adenomyosis or adenomyoma, treatment of adenomyosis or adenomyoma with office hysteroscopy. Results: The literature showed that hysteroscopy can identify superficial adenomyosis. There are a variety of hysteroscopic images that can be connected with the disease. New equipment like the spirotome has been used to access deeper layers of myometrium and obtain biopsies under direct vision from the adenomyotic areas. Different methods of treatment have been also described, like enucleation of focal superficial adenomyoma, coagulation, evacuation of cystic adenomyosis when the lesion is smaller than 1.5 cm, and resection of adenomyotic nodules in case of bigger lesions (>1.5 cm). Diffuse superficial adenomyosis is also managed by resection. Conclusions: Hysteroscopy has revolutionised the approach to adenomyosis. It is a useful tool in assessing mainly superficial adenomyosis. The role of hysteroscopy in surgical management of adenomyosis need to be confirmed with further studies.

## 1. Introduction

Adenomyosis is a chronic disease, known more than 100 years. The incidence of adenomyosis can only be estimated between 5–70%, since it is a disease whose diagnosis can be confirmed only after hysterectomy [1]. Histologically, adenomyosis is characterised by the invasion of myometrium from endometrial glands and/ or stroma deeper than 2.5 mm from the endometrial/ myometrial junction (junction zone-JZ) [2]. 

In most cases, adenomyosis does not present with specific clinical symptoms; instead, it causes dysmenorrhea starting at an early age around the time of menarche and chronic pelvic pain, which are non-responsive to analgesics or cyclic oral contraceptives [3]. Most patients with adenomyosis complain of abnormal uterine bleeding (AUB), abdominal bloating, and low pelvic pressure. Regardless of clinical manifestation, one out of three women with adenomyosis remains asymptomatic [4]. In many cases, adenomyosis co-exists with endometriosis, while only a small fraction of patients with endometriosis do not present with concurrent adenomyosis [5].

The diagnosis of adenomyosis is still controversial. In this study, we are aiming to explore the diagnostic and therapeutic role of office hysteroscopy in adenomyosis to identify types of adenomyosis that are suitable for hysteroscopic diagnosis and potential treatment in an office set-up.

## 2. Classification and Diagnosis of Adenomyosis

Adenomyosis can be distinguished in a histologic specimen as diffuse or focal according to the myometrial invasion. In the diffuse type, endometrial glands and/or stroma are extensively found between myometrial muscle fibres, which leads to an increase in uterine volume resulting in the pathognomonic “bulky” uterus, and traditionally can be detected during clinical examination. Focal adenomyosis is generally a single solid nodule located in the myometrium, which is characterised as adenomyoma, or an adenomyotic cyst when appearing with cystic characteristics [6].

Another classification of adenomyosis is based on magnetic resonance imaging (MRI). The evaluation of the junctional zone (JZ) is based on three main parameters: thickness, regularity, and interruption of JZ. Ιrregularity is measured as JZdif and it is considered an additional characteristic, suggestive of adenomyosis [7]. Kishi et al. [8] created a different classification based on MRI characteristics. Subtype I (intrinsic) adenomyosis is related with inner structural components of the uterus, such as the endometrium and the junctional zone. Sub-type II (extrinsic) adenomyosis arose in the outer shell of the uterus, disrupting the serosa but not affecting the inner components. Subtype III (intramural) adenomyosis resided solitarily in the myometrium. The remainder of images conclude to an additional subtype, subtype IV (indeterminate) adenomyosis.

Kobayashi et al. [9] used a different classification for adenomyosis based on the affected area and extent of adenomyosis. More specifically, patients were identified with three types of adenomyosis, intrinsic, extrinsic, and other types. The intrinsic type is defined as adenomyosis present in the uterine inner layer without affecting the outer structures of myometrium. The extrinsic type is defined as adenomyosis that occurs in the uterine outer layer without affecting the inner structures. The extent of adenomyosis lesion is further categorized into three volumes (<1/3, <2/3, or >2/3 of uterine wall). A1, A2 and A3 are defined as “the lesion is confined to the inner 1/3 of the uterine myometrium”, “the lesion is confined to the inner 2/3 of the uterine myometrium”, and “the lesion extends beyond the inner 2/3 of the myometrium and part of the lesion reaches the uterine serosa.” B1, B2, and B3 are defined as “the lesion is confined to the outer 1/3 of the uterine myometrium”, “the lesion is confined to the inner 2/3 of the uterine myometrium”, and “the lesion extends beyond the outer 2/3 of the myometrium and part of the lesion reaches the uterine endometrium.” If the lesion extends to the entire myometrium, A3 and B3 are indistinguishable using MRI and pathology. Patients can also be classified as “other type” when they did not belong to either type A or type B.

Women with adenomyosis are not presented with the same clinical picture. In addition, for some of them it is necessary to keep their uterus. As result, there is a need to identify adenomyosis pre-operatively in order to decide the best treatment and the right surgical technique and achieve better results for each patient [10]. Nowadays, it is more feasible than in the past to diagnose adenomyosis with hysteroscopy, transvaginal sonography (TVS), and magnetic resonance imaging (MRI), without proceeding to hysterectomy [11]. MRI used to be the optimal modality to detect adenomyosis [12], but it is not a cost effective option and is not always available [13]. TVS is a more feasible option and its accuracy in diagnosing adenomyosis has been increased significantly following the improvements of both sonographic hardware and software [14,15]. The direct visualization of the uterine cavity offered by hysteroscopy has been suggested to broaden the possibilities of diagnosing adenomyosis, offering at the same time some interventional properties [16,17].

Hysteroscopic techniques and equipment have changed throughout the years, making it available in an ambulatory setting. Initially, hysteroscopy took place only in theatre under general analgesia, but since then, hysteroscopy has been introduced in an office setting, where no anaesthesia nor analgesia is used, mainly attributed to the minimization of the equipment and the introduction of the new, ‘no-touch’ technique called vaginoscopy [18]. Having hysteroscopy performed in an office set-up gives the possibility of providing a “see and treat” option, so the hysteroscopist can investigate, diagnose, and in several cases, treat the endometrial pathology in a single appointment [18].

## 3. Material and Methods

Material and Methods: The literature was searched via Pubmed and Embase with the following headings: diagnosis of adenomyosis or adenomyoma and office hysteroscopy, hysteroscopy findings of adenomyosis or adenomyoma, treatment of adenomyosis or adenomyoma with office hysteroscopy. Studies were included only if they presented data about office hysteroscopy and its application in cases of adenomyosis, including diagnostic only or operative hysteroscopy too.

## 4. Hysteroscopic Diagnosis of Adenomyosis

It is difficult to determine the role of hysteroscopy in identifying adenomyosis. A hysteroscopist is able to visualise the endometrial cavity in real time and also to obtain endometrial biopsies under direct vision, increasing the patient’s safety and minimizing false negatives [16]. However, there are still no specific characteristics that can safely define the diagnosis of adenomyosis [19]. An important limitation of the technique is that hysteroscopy allows the observation of the endometrial surface only. As a result, adenomyosis that is usually found at the deeper layer of myometrium is impossible to identify [5,10].

The following hysteroscopic patterns are generally considered indicative of adenomyosis [20]:Irregular endometrium with tiny openings seen on the endometrial surface (Figure 1)Hypervascularisation (Figure 2)An endometrial “strawberry” pattern (Figure 3)Fibrous cystic appearance of intrauterine lesions (Figure 4)Haemorrhagic cystic lesions presenting with a dark blue or chocolate brown appearance (Figure 5)

In patients with adenomyosis, hysteroscopy alone can provide an impression of possible adenomyotic lesions, with a pattern of irregular vascular distribution, during proliferative and secretory phase, without being able to set a diagnosis at times. However, it allows direct biopsy samples obtained from the endometrium and underlying myometrium using mechanical instruments (biopsy or grasping forceps, scissors) or bipolar electrodes [16,21].

In the presence of any hysteroscopic pattern suggestive of the disease, the surgeon can use the resectoscope with a diathermic loop to perform resection of the endo-myometrial layer. To obtain an adequate biopsy with this approach, it is necessary to obtain specimens from both the endometrium and myometrium, which facilitates accurate histologic diagnosis. For the aforementioned reason, a second biopsy should follow the initial one so that deeper specimens can be obtained in order to include only myometrial tissue [6]. The following three findings upon biopsy with resectoscope are suggestive of adenomyosis: (1) irregular subendometrial myometrium (spiral and/or fibrotic); (2) irregular myometrial architecture during resection; (3) presence of intramural endometriomas [6]. The use of a mini-resectoscope will potentially allow the obtaining of biopsies in cases of adenomyosis in an office set-up because of its size, which make the experience of office hysteroscopy more comfortable for patients [20].

Vercellini et al. [22] suggested that needle biopsy provides high specificity (95%), though with low sensitivity (44.8%). Even when needle biopsy was combined with the application of transvaginal ultrasound, there was no improvement in sensitivity. Darwish et al. proved that the resectoscopic technique is superior to the rigid biopsy forceps [23], while McCausland was the first to describe the technique of myometrial biopsy through hysteroscopy [17]. He diagnosed adenomyosis in 66% of his cases and found a corelation between the depth and clinical symptoms. Goswami et al. [24], using the loop resection, found an incidence of adenomyosis in 60%, which was in contrast to the control group with 33%. The mean depth of invasion was 4.103 mm in the menorrhagia group and 2.03 mm in the control group. Fedele et al. [25] used, as a threshold, 2.5 mm and diagnosed adenomyosis in 51% of cases. Though there is no consensus with regard to the diagnostic hysteroscopic criteria of adenomyosis, it seems though that hysteroscopy can play an important role, being able to diagnose this pathology. When the use of needle biopsy is compared with the use of resectoscope as diagnostic tool, the second is preferred [24].

Gordts et al. described the use of an alternative system, the Trophy Hysteroscope (Karl Storz, Germany). This system can change from diagnostic, with a 2.9 mm diameter hysteroscope, to an operative, with a 4.4 mm scope, without the need to withdraw the hysteroscope [21]. Further 5-Fr instruments can be inserted through the operative sheath and used for dissection or coagulation. Endo-myometrial biopsy showed a specifity of 78.46% with a low sensitivity of 54.32%, the latter mostly related to the high amount of false negatives in the cases of deep adenomyosis [26]. On the contrary, ultrasound has a sensitivity of 72% [27]. The use of utero-spirotome offers a deeper and more direct biopsy from both endometrium and myometrium at the same time. The spirotome operates with two devices: the receiving needle with a cutting helix at the distal end, and a cutting cannula as an outer sheet; the direction and position of the helix point must be under continuous ultrasonographic and hysteroscopic guidance [21]. Due to its suboptimal size, the use of the above mentioned hysteroscope is considered only under general anaesthesia, limiting its use to the operating theatres.

Dakhly et al. presented an hysteroscopic tissue sampling approach for the diagnosis of adenomyosis with subsequent confirmatory hysterectomy pathology. A hysteroscopic biopsy was taken from the posterior uterine wall using only hysteroscopic scissors and graspers. This method showed a 54.3% sensitivity and a 78.5% specificity for diagnosing adenomyosis amongst 292 premenopausal women who were meant to have a hysterectomy for dysmenorrhea and heavy menstrual bleeding symptoms [26].

There is a large range of sensitivity amongst tissue sampling techniques during hysteroscopy. Most of the papers published report a suboptimal and low sensitivity for the diagnosis of adenomyosis. It is possible, though, since the number of biopsies, the location of the biopsy, and the optimal biopsy technique are all potential factors that define the sensitivity in hysteroscopic diagnosis of adenomyosis. The heterogeneity between the techniques presented in literature it makes it impossible to compare them and draw unbiased results [28].

Hysteroscopy provides the possibility of direct biopsies aiming at the areas of interest. These results from hysteroscopy are corelated with ultrasound images. Clinicians can approach women with adenomyosis differently, more conservatively, and without the necessity of hysterectomy [29] to set the definite diagnosis. It has to be noted that evidence is not enough to support the application of hysteroscopy, and particularly office hysteroscopy, as method of investigation and diagnosis of adenomyosis. There are not many studies that support the use of intrauterine modalities, like hysteroscopy, in the diagnosis of adenomyosis [30]. Nevertheless, hysteroscopy is a less invasive techniques when is compared with hysterectomy, and it is clearly necessary to fully explore its role in adenomyosis.

## 5. Office Hysteroscopy as a Treatment Option for Adenomyosis

Adenomyosis can be approached conservatively or surgically. Traditionally the most common way to treat this chronic disease is hysterectomy. Pre-operative diagnosis of adenomyosis has been improved significantly, such as the need of offering tailored treatment depending on the patient’s age, desire for a future pregnancy, and symptomatology [31].

Regarding medical management, the available options are primary, aiming to alleviate local hyperestrogenism and improve the most severe symptoms of adenomyosis, such as heavy menstrual bleeding, dysmenorrhea, and non-menstrual related chronic pelvic pain. Medical treatments used for adenomyosis include gonadotropin-releasing hormone agonists or antagonists, levonorgestrel-releasing intrauterine device (LNG-IUD), oral contraceptive combined pill, progestogens, Ulipristal Acetate, and non-steroidal inflammatory drugs (NSAID’s); modalities similar to these are in use for the treatment of endometriosis [31,32]. The principal limit of these medical options is that they induce regression rather than eradication of the pathology, with symptoms’ recurrence after drug discontinuation [33].

Focal adenomyosis is excised by laparoscopy or laparotomy in a way similar to myomectomy. However, it can be challenging at times, since adenomyosis occasionally lacks a clear cleavage plane and there is a risk of the clinician excising more tissue than it is necessary [26]. Hysteroscopy is feasible only in occasions where focal or diffuse adenomyosis is found close to the endometrium [20]. Hysteroscopy is not considered to be the first choice of treatment for women with adenomyosis, though there are some cases to prove that hysteroscopy can play an important role in the management of adenomyosis [19,20].

Moreover, Di Spezio et al. [20] suggested enucleation as technique for focal adenomyomas less than 1.5 cm in diameter and close to endometrial cavity, using mechanical instruments and/or bipolar electrodes. This treatment is applicable in cases of visible areas of adenomyosis with hysteroscopy, as they expand into the endometrial cavity. The rationale behind this is that adenomyosis can be excised with a minimally invasive dissection and can take place in an office setting, by using the mini-hysteroscope and resectoscope. The technique used in ambulatory setting is the same technique used for enucleation of submucosal myomas presenting with intramural components. However, the procedure involves a considerable element of precautionary exploration due to the lack of a distinct cleavage plane required for adequate identification of healthy myometrial tissue [16]. In case of an adenomyotic nodule of more than 1.5 cm, resectoscopic treatment could be a feasible treatment option [20]. Resectoscopic excision is possible in an office set-up when a mini-resectoscope is used [34]. The evidence, though, is limited as regards the use of this operative technology and treatment of adenomyosis.

Endometrial ablation is suggested mainly to women not desiring future pregnancy, and when it is performed, the underlying defective myometrium is removed [34]. In this case, it is important to identify women that could potentially complete the procedure in an office setting, due to the additional time requirements as well as the discomfort that this can cause [34]. In cases of deep diffuse adenomyosis, where the adenomyotic lesions are deeper than the superficial layer of the endometrium, hysteroscopy is not considered the appropriate approach, as it is expected. McCausland et al. showed that in these cases, resectoscopic treatment cannot ameliorate the complaining symptoms, and moreover may hide deep adenomyosis by allowing persistent development of the disease below the endo-myometrial scar tissue [35].

Operative hysteroscopy may be suitable for cases of superficial adenomyotic nodules and for diffuse superficial adenomyosis as a treatment modality [20]. When an adenomyoma is visible in the uterine cavity, the technique of adeno-myomectomy can be employed in a way similar to myomectomy [36]. Xia et al. [37] presented data of 51 women that underwent hysteroscopic resection of adenomyotic lesions under ultrasound guidance and proved the clinical feasibility of hysteroscopic resection to treat symptomatic adenomyosis lying in the deeper layers of the myometrium. The intraoperative and postoperative safety and low recurrence rate of menorrhagia and dysmenorrhea during the 2-year follow-up demonstrated the potential applications of hysteroscopy in treating adenomyosis, as previously reported [38,39]. Unfortunately, hysteroscopy only provides information for the endometrial lining, and the hysteroscopist faces significant limitations with regard to cleavage plane and space. Moreover, muscular tissue is thin and easily perforated; therefore, it is vital to carefully select patients and ensure that this surgical intervention is carried out by a skilled surgeon [37]. More specifically, Xia et al. [37] presented the technical steps of removing adenomyomas with the use of hysteroscopic resection. Upon cutting the endometrium which covers the adenomyotic lesions, pink ectopic endometrial lesions in the myometrium are exposed. The ectopic endometrium and adenomyotic lesions in the myometrium are then gradually resected. During the resection of lesions, several intramural microcysts with wide bases are revealed. Opening the microcyst results in the outflow of a thick brownish fluid, mostly old blood. Then, the endometrium-like tissue and microcysts are resected with the use of a loop. During resection, the myometrial vessels are coagulated to avoid excessive fluid absorption. After fluid drainage, an internal view of the microcyst reveals the presence of pink ectopic endometrium-like tissue. The operation is considered complete when the pink fasciculate structure of the myometrium appears [37]. Because of the nature of the procedure, it is thought not to be safe to be performed with office hysteroscopy equipment [37].

Superficial diffuse adenomyosis may be also treated with endo-myometrial ablation [20]. This approach is quite different to the traditional method of endometrial ablation. Resection must expand further into the myometrium, at least 10 mm in depth, while the adenomyotic lesion has to be between 3–10 cm. Then, the hysteroscopist will need to continue cutting the myometrial layer until healthy myometrium is recognised. Coagulation will conclude the procedure, which is completed using 3 mm or 5 mm straight loops for ablation of the fundus and classical cutting loops for ablation of the uterine walls [20]. When symptoms persist, a second-look surgical procedure can take place to assess the efficacy of the hysteroscopic technique when treating adenomyosis [20]. Data is not available to evaluate this method for an office set-up.

Preutthipan et al. [32] presented their results after hysteroscopic rollerball endometrial ablation. Of the women who participated in that study, 98.4% reported decreased bleeding. Only 1.6% of patients had to proceed with hysterectomy to treat persisting symptoms. This study proved that hysteroscopic rollerball endometrial ablation, as a surgical alternative to the excision of adenomyosis, is an effective and safe procedure. Furthermore, it can also reduce the need for unnecessary major interventions such as hysterectomy. Nevertheless, the main limitation of the aforementioned study was the limited sample size and its retrospective character. This study also presents the results from a hysteroscopy performed in theatre under general anaesthesia [32].

When cystic lesions are pre-operatively recognised deeper in the myometrium, spirotome might be a efficient approach [19]. Gordts et al., showed that by using spirotome under ultrasound guidance, adenomyotic cystic areas can be addressed, even in cases where intracavitary components are not visible [19]. The device creates a channel and provides hysteroscopic access to the cystic structure, allowing further treatment with bipolar coagulation or resection to follow [21]. Mechanical dissection with small scissors or cystic opening and coagulation of small adenomyotic cystic walls and crypts in the junctional zone can be performed under ultrasound guidance too, although this procedure is only described in some cases [40,41,42,43].

As has been reported earlier, there is not a globally accepted method to hysteroscopically treat adenomyosis. Some authors are presenting their own experience. In this study we are aiming to present these different methods. We are not able, though, to prove which one is the best because of their heterogeneity, even in the principles and data given in each manuscript.

## 6. Conclusions

Hysterectomy will always be the standard treatment of adenomyosis, but an increasing number of women are willing to proceed with fertility-sparing approaches. In this case, it is possible to offer medical management of the disease, or a hysteroscopic approach when it is indicated in selected patients. An improvement of dysmenorrhea and menorrhagia is achieved in more than 81% and 50% of patients, respectively [20]. It seems that conservative management with medical treatment is more effective regarding adenomyosis, but the evidence is still limited to support its superiority with regard to reproductive outcome [20]. This highlights the importance of on-going study on the impact of hysteroscopic treatment in cases of adenomyosis.

Hysteroscopy seems to offer additional information in understanding adenomyosis as a pathologic condition, providing diagnosis and treatment for specific types of the disease. When adenomyosis is close to the endometrium, there are hysteroscopic features that can be detected, providing extra information regarding the uterine cavity of affected patients [20]. It offers the additional benefit of obtaining biopsies under direct vision, or with ultrasound guidance for cases of deeper adenomyosis, to confirm the endoscopic diagnosis [5]. Diagnostic hysteroscopy in cases of adenomyosis can take place in an office set-up with optimal results [20]. General or local anaesthesia is not necessary, and the hysteroscopist is able to employ instruments with a smaller diameter and efficiently approach the endometrial cavity so that biopsies under direct vision can be easily obtained [20].

Conservative, uterine-sparing treatments of adenomyosis appear to be feasible and efficacious. Hysteroscopy can offer an option of treatment for adenomyosis. Further research is necessary to better understand the indication of hysteroscopic treatment of adenomyosis and its impact in womens’ lives [20], especially in an office environment, since available information so far is not adequate to support the treatment of adenomyosis with office hysteroscopy [39,40].

## Figures and Tables

**Figure 1 diagnostics-13-02182-f001:**
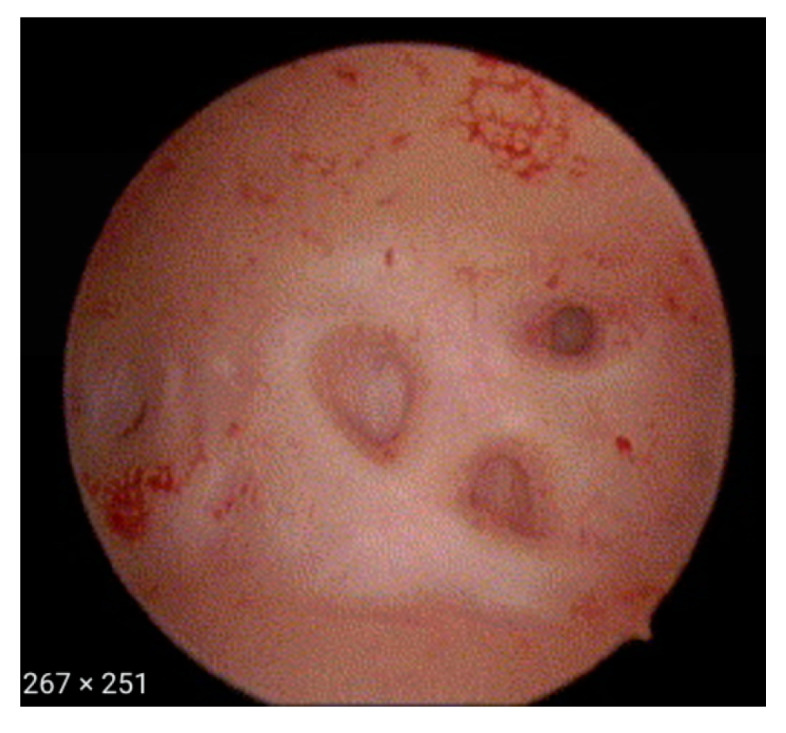
Irregular endometrium with openings.

**Figure 2 diagnostics-13-02182-f002:**
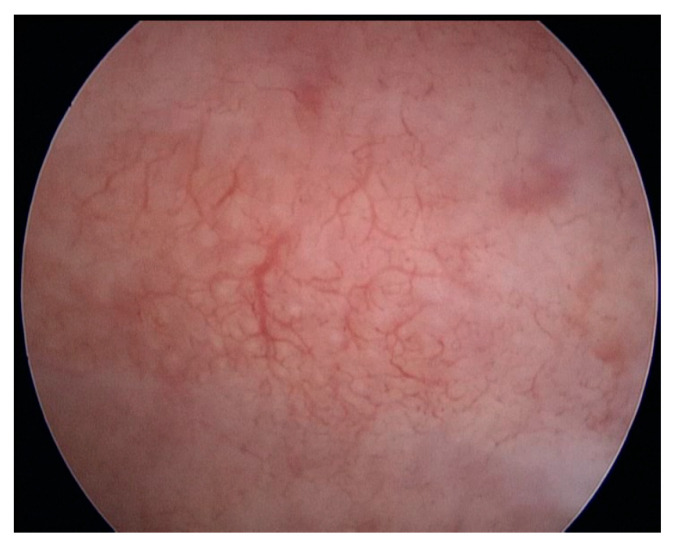
Hypervascularisation.

**Figure 3 diagnostics-13-02182-f003:**
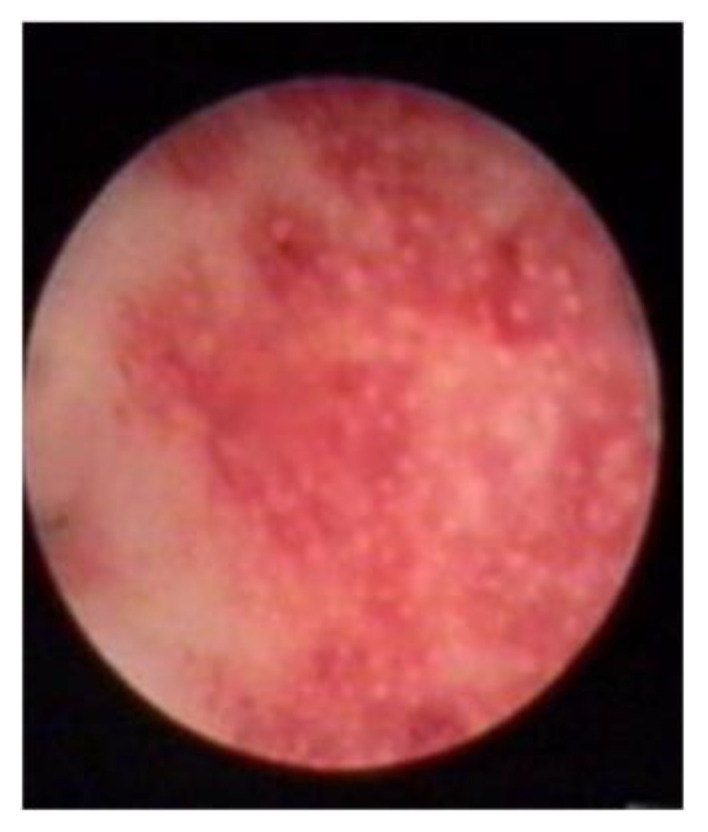
An endometrial “strawberry” pattern.

**Figure 4 diagnostics-13-02182-f004:**
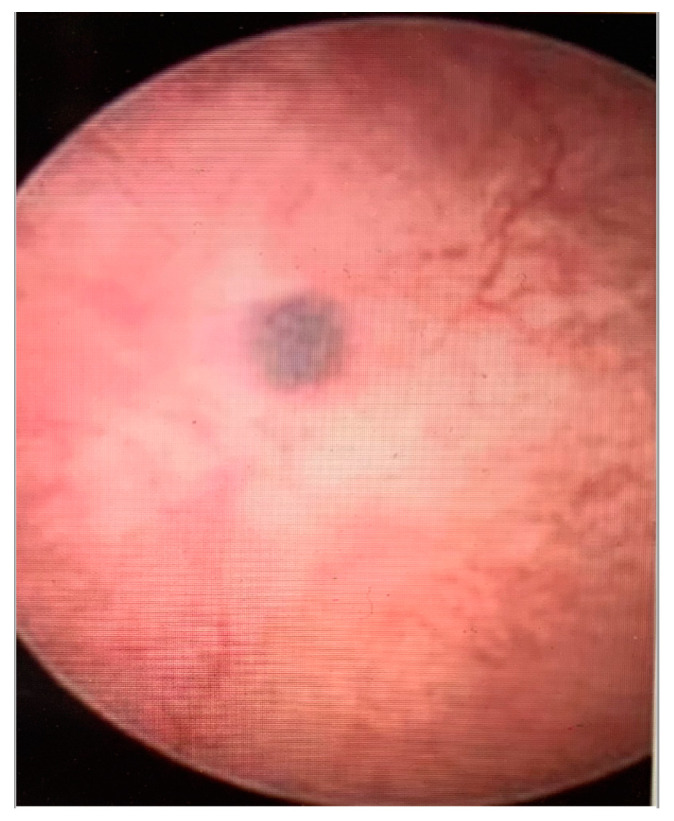
Fibrous cystic appearance of intrauterine lesions.

**Figure 5 diagnostics-13-02182-f005:**
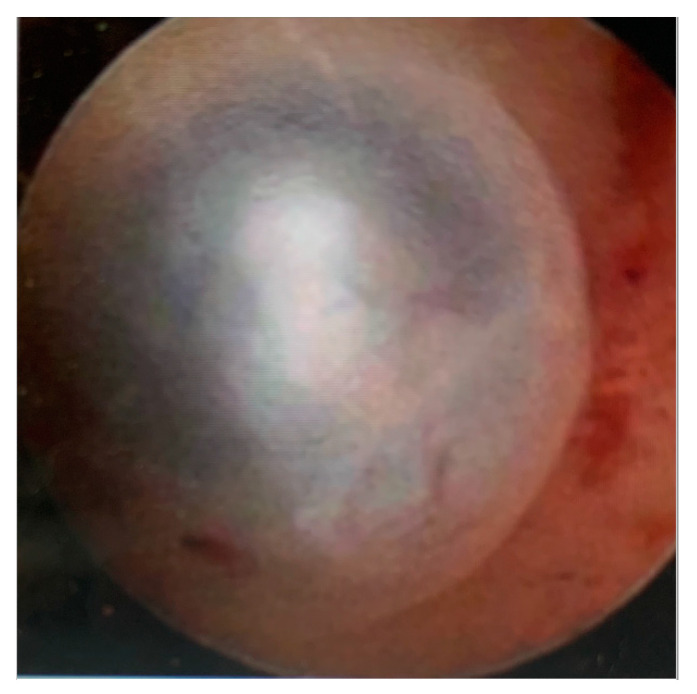
Haemorrhagic cystic lesions presenting with a dark blue or chocolate brown appearance.

## Data Availability

Not applicable.

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
