# Peer review of "Diagnosis and Treatment of Adenomyosis with Office Hysteroscopy—A Narrative Review of Literature"

_diagnostics, 2023, doi:10.3390/diagnostics13132182_

Round 1
Reviewer 1 Report
This manuscript is a very useful review that summarizes the knowledge of much of the literature that hysteroscopy is a useful option for the diagnosis and treatment of adenomyosis. However, it only addresses the relatively old method of classifying subtypes of uterine adenomyosis, which divides the disease into diffuse and focal subtypes. Although many researchers have classified uterine adenomyosis into their own subtypes, the classification of Kishi et al.1) has been used more frequently in recent years, both clinically and in research. The present manuscript mainly describes type 1 and 3 of the Kishi classification, with little discussion of type 2 (or the extrinsic type of Kobayashi et al.2). In this subtype, the lesions spread from the serosal surface of the uterus to the lumen, so the endometrium is less affected, and the findings on hysteroscopy are considered to be poor. However, it would not be a review of adenomyosis in general without a description of this type. Additional mention of it should be made.
Furthermore, the conclusion is too long, perhaps because it uses the same description method as the main text. Normally, references should not be cited in the conclusion. The conclusions should be more concise.
Once they have been corrected, this manuscript will be considered accepted.
Other comments are listed below.
1.INTRODUCTION: The facts are too elementary to be included even as an introduction to a review of uterine adenomyosis. It is intended for readers with little or no knowledge of adenomyosis. Rather, it should be more centered on why this review was written. It is not a textbook.
2.  RESULTS: It seems to be very well organized, but the citations in Ref. 22 are conspicuous. The fact that it cites a lot from the content is unavoidable, but it feels like a summary manuscript of Ref. 22.
3.RESULTS: Page 7, lines 173-174 "The principal limit of these~recurrence after drug discontinuation.": this fact should also be cited, see article Matsushima et al 3).
1)Kishi Y, Suginami H, Kuramori R, Yabuta M, Suginami R, Taniguchi F. Four subtypes of adenomyosis assessed by magnetic resonance imaging and their specification. Am J Obstet Gynecol. 2012 Aug;207(2):114.e1-7. doi: 10.1016/j.ajog.2012.06.027. Epub 2012 Jun 19. PMID: 22840719.
2)Kobayashi H, Matsubara S, Imanaka S. Clinicopathological features of different subtypes in adenomyosis: Focus on early lesions. PLoS One. 2021 Jul 14;16(7):e0254147. doi: 10.1371/journal.pone.0254147. PMID: 34260636; PMCID: PMC8279363.
3)Matsushima T, Akira S, Yoneyama K, Takeshita T. Recurrence of uterine adenomyosis after administration of gonadotropin-releasing hormone agonist and the efficacy of dienogest. Gynecol Endocrinol. 2020 Jun;36(6):521-524. doi: 10.1080/09513590.2019.1683818. Epub 2019 Oct 29. PMID: 31661345.
Author Response
Point 1. This manuscript is a very useful ………….. Additional mention of it should be made.
Kishi et al (9) create a different classification based on MRI characteristics. Subtype I (intrinsic) adenomyosis had is related with inner structural components of the uterus, such as the endometrium and the junctional zone. Sub-type II (extrinsic) adenomyosis arose in the outer shell of the uterus disrupting the serosa but not affecting the inner components. Subtype III (intramural) adenomyosis resided solitarily in the myometrium. The remainder of images conclude to an additional subtype, subtype IV (indeterminate) adenomyosis.
Kobayashi et al (9) used a different classification for adenomyosis based on the affected area and extent of adenomyosis. More specifically, patients were identified with three types of adenomyosis, intrinsic, extrinsic, and other types. The intrinsic type is defined as adenomyosis present in the uterine inner layer without affecting the outer structures of myometrium. The extrinsic type is defined as adenomyosis that occurs in the uterine outer layer without affecting the inner structures. The extent of adenomyosis lesion is further categorized into three volumes (<1/3, <2/3, or >2/3 of uterine wall). A1, A2 and A3 are defined as "the lesion is confined to the inner 1/3 of the uterine myometrium", "the lesion is confined to the inner 2/3 of the uterine myometrium", and "the lesion extends beyond the inner 2/3 of the myometrium and part of the lesion reaches the uterine serosa." B1, B2, and B3 are defined as "the lesion is confined to the outer 1/3 of the uterine myometrium", "the lesion is confined to the inner 2/3 of the uterine myometrium", and "the lesion extends beyond the outer 2/3 of the myometrium and part of the lesion reaches the uterine endometrium." If the lesion extends to the entire myometrium, A3 and B3 are indistinguishable by MRI and pathology. Patients can also be classified as "other type" when they did not belong to either type A or type B.
Point 2. Furthermore, the conclusion is too long, perhaps because it uses the same description method as the main text. Normally, references should not be cited in the conclusion. The conclusions should be more concise.
Conclusion aims to summarise each subject mentioned at the main manuscript. This is the reason why it is 3 paragraphs and also references are mentioned.
Point 3. INTRODUCTION: The facts are too elementary to be included even as an introduction to a review of uterine adenomyosis. It is intended for readers with little or no knowledge of adenomyosis. Rather, it should be more centered on why this review was written. It is not a textbook.
Adenomyosis is a chronic disease, known more than 100 years. It was Karl Freiherr von Rokitansky that described first adenomyosis in German literature as ‘‘fibrous tumours containing gland like structures that resemble endometrial glands’’ (1). Cullen characterised adenomyosis as ‘‘endometriosis with predominantly presence of fibromuscular tissue’’ in 1921(2). The incidence of adenomyosis can only be estimated between 5%-70%, since it is a disease that its diagnosis can be confirmed only after hysterectomy (3). Histologically, adenomyosis is characterised by the invasion of myometrium by endometrial glands and/ or stroma, deeper than 2.5mm from the endometrial/ myometrial junction (junction zone-JZ) (4).
Adenomyosis can be distinguished in a histologic specimen, as diffuse or focal according to the myometrial invasion. In the diffuse type, endometrial glands and/or stroma are extensively found between myometrial muscle fibres, which leads to an increase in uterine volume resulting in the pathognomonic “bulky” uterus, and traditionally can be detected during clinical examination. Focal adenomyosis is generally a single solid nodule located in the myometrium, which is characterised as adenomyoma or an adenomyotic cyst, when appeared with cystic characteristics (5).
Another classification of adenomyosis is based on Magnetic Resonance Imaging (MRI). The evaluation of the junctional zone (JZ) is based on three main parameters: thickness, regularity, and interruption of JZ. Diffuse adenomyosis is diagnosed given a JZmax of 15mm, while a borderline JZ of 12 mm to 15 mm in thickness, requires the presence of additional criteria (such as loss of smooth appearance of the JZ, poorly circumscribed foci within the myometrium with high or low intensity) in order to set the diagnosis (6,7). In cases of focal adenomyosis, T2-weighted sequences show a decreased subendometrial signal intensity (consistent with the presence of necrotic tissue) with blurred margins surrounding the lesion (6). An additional parameter is the difference between JZmax and JZmin , known as JZdif (JZ difference), calculating the distance between the anterior and the posterior border. Ιrregularity is measured as JZdif and it is considered an additional characteristic, suggestive of adenomyosis (8).
Kobayashi et al (9) used a different classification for adenomyosis based on the affected area and extent of adenomyosis. More specifically, patients were identified with three types of adenomyosis, intrinsic, extrinsic, and other types. The intrinsic type is defined as adenomyosis present in the uterine inner layer without affecting the outer structures of myometrium. The extrinsic type is defined as adenomyosis that occurs in the uterine outer layer without affecting the inner structures. The extent of adenomyosis lesion is further categorized into three volumes (<1/3, <2/3, or >2/3 of uterine wall). A1, A2 and A3 are defined as "the lesion is confined to the inner 1/3 of the uterine myometrium", "the lesion is confined to the inner 2/3 of the uterine myometrium", and "the lesion extends beyond the inner 2/3 of the myometrium and part of the lesion reaches the uterine serosa." B1, B2, and B3 are defined as "the lesion is confined to the outer 1/3 of the uterine myometrium", "the lesion is confined to the inner 2/3 of the uterine myometrium", and "the lesion extends beyond the outer 2/3 of the myometrium and part of the lesion reaches the uterine endometrium." If the lesion extends to the entire myometrium, A3 and B3 are indistinguishable by MRI and pathology. Patients can also be classified as "other type" when they did not belong to either type A or type B.
In most cases, adenomyosis does not present with specific clinical symptoms, instead it causes dysmenorrhea starting at an early age around the time of menarche and chronic pelvic pain, which are not responding to analgesics or cyclic oral contraceptive (10). Most patients with adenomyosis complain of abnormal uterine bleeding (AUB), abdominal bloating and low pelvic pressure. Interestingly although multiparity and iatrogenic causes such as dilatation and curettage (D&C) are considered major risk factors for the development of the disease, other symptoms related with adenomyosis are recurrent miscarriages and morbidity adherent placenta negatively affecting reproductive outcome. Regardless of clinical manifestation, one out of three women with adenomyosis remains asymptomatic (11). In many cases, adenomyosis co-exists with endometriosis, while only a small fraction of patients with endometriosis do present with concurrent adenomyosis (12).
Women with adenomyosis are not presented with the same clinical picture. In addition, for some of them is necessary to keep their uterus. As result, there is a need to identify adenomyosis pre-operatively, in order to decide the best treatment, the right surgical technique and achieve better results for each patient (13). Nowadays, it is more feasible than in the past to diagnose adenomyosis with hysteroscopy, transvaginal sonography (TVS) and Magnetic Resonance Imaging (MRI), without proceeding to hysterectomy (14). MRI used to be the optimal modality to detect adenomyosis (15) , but it is not a cost effective option and is not always available (16). TVS is a more feasible option and its accuracy in diagnosing adenomyosis has been increased significantly, following the improvements of both sonographic hardware and software (17,18) . The direct visualization of the uterine cavity, offered by hysteroscopy, has been suggested to broaden the possibilities of diagnosing adenomyosis, offering at the same time some interventional properties (19,20).
Hysteroscopic technique and equipment has changed throughout the years making it available in an ambulatory setting. Initially, hysteroscopy was taking place only in theatre under general analgesia, but since then, hysteroscopy has been introduced in an office setting, where no anaesthesia nor analgesia is used, mainly attributed to minimization of the equipment and the introduction of the new, ‘no-touch’ technique called vaginoscopy (21). Having hysteroscopy performed in an office set up gives the possibility to provide “see and treat” option, so the hysteroscopist can investigate, diagnose and in several cases treat the endometrial pathology in a single appointment (21).
Diagnosis of adenomyosis is still controversial. In this study, we are aiming to explore the diagnostic and therapeutic role of office hysteroscopy in adenomyosis, to identify types of adenomyosis that are suitable for hysteroscopic diagnosis and potential treatment in office set up.
Point 4. RESULTS: It seems to be very well organized, but the citations in Ref. 22 are conspicuous. The fact that it cites a lot from the content is unavoidable, but it feels like a summary manuscript of Ref. 22.
I have modified and replace references.
Point 5: RESULTS: Page 7, lines 173-174 "The principal limit of these~recurrence after drug discontinuation.": this fact should also be cited, see article Matsushima et al
The principal limit of these medical options is that they induce regression rather than eradication of the pathology, with symptoms’ recurrence after drug discontinuation (36).
References
All three suggested references have been added.
Reviewer 2 Report
Excellent review on the use of hysteroscopy in the diagnosis and treatment of adenomyosis.
I would just like to point out that spaces are missing in some paragraphs.
Page 7 line 202 Xia et al(38)
Page 9 line 280 100kg and
Page 9 line 281 5cm, while adenomyotic lesions have to be of 3-10cm
Author Response
Page 7 line 202 Xia et al(38)
Page 9 line 280 100kg and
Page 9 line 281 5cm, while adenomyotic lesions have to be of 3-10cm
all these comments have been taken into consideration and added accordingly at the manuscript in yellow.